# Genome-Wide Characterization of Soybean Hexokinase Genes Reveals a Positive Role of *GmHXK15* in Alkali Stress Response

**DOI:** 10.3390/plants12173121

**Published:** 2023-08-30

**Authors:** Feng Jiao, Yang Chen, Dongdong Zhang, Jinhua Wu

**Affiliations:** College of Agriculture, Heilongjiang Bayi Agricultural University, Daqing 163319, China; fengjiao2021@126.com (F.J.); lbt22222@163.com (Y.C.); a230224127@126.com (D.Z.)

**Keywords:** hexokinase, expression, enzyme activity, abiotic stress, soybean (*Glycine max* L.)

## Abstract

Hexokinase (HXK) proteins catalyze hexose phosphorylation and are important for the sensing and signaling of sugar. In order to determine the roles played by HXKs in soybean growth and stress responsiveness, seventeen HXK genes (*GmHXK1-17*) were isolated and analyzed. The phylogenic analysis and subcellular location prediction showed that GmHXKs were clearly classified into type A (GmHXK1-4) and type B (GmHXK5-17). There were similar protein structures and conserved regions in GmHXKs to the HXKs of other plants. An expression analysis of the *GmHXK* genes in soybean organs or tissues demonstrated that *GmHXK3* and *GmHXK12*, *15*, and *16* were the dominant HXKs in all the examined tissues. In addition, salt, osmotic, and alkaline stress treatments dramatically increased the activity and transcripts of *GmHXKs*. There is the possibility that a type-B isoform (GmHXK15) plays a crucial role in soybean adaptation to alkali, as the expression levels of this isoform correlate well with the HXK enzyme activity. Based on an enzyme assay performed on recombinant plant HXK15 proteins expressed in *Escherichia coli*, we found that GmHXK15 had functional HXK activities. A further analysis indicated that GmHXK15 specifically targeted the mitochondria, and the overexpression of the *GmHXK15* gene could significantly enhance the resistance of transgenic soybean to alkali stress. The present findings will serve as a basis for a further analysis of the function of the *GmHXK* gene family.

## 1. Introduction

In addition to providing carbon and energy for growth and development, sugars play a critical role in gene-expression regulation across a wide range of critical processes, such as photosynthesis, hormone biosynthesis, signaling, glyoxylate metabolism, carbon and nitrogen metabolism, pathogen defense, stress response, respiration, and senescence [1,2]. HXK catalyzes the phosphorylation of glucose in the glycolytic pathway and contributes to the production of nucleotide sugars required for the biosynthesis of the cell wall and secondary metabolic pathways, in addition to the oxidative pentose phosphate pathway [3]. Evidence from recent studies suggests that HXK can function as a sugar sensor by interacting with several signaling pathways to regulate plant growth and development [4,5]. The HXKs gene family, which plays an important role in plant growth and development, has been extensively characterized through functional, biochemical, and computational approaches in a wide range of plant species, including *Arabidopsis thaliana* [6], *Physcomitrella patens* [7], *Brassica napus* [8], *Lycopersicon esculentum* [9,10], *Solanum Lycopersicum* [11], *Zea mays* [12], *Nicotiana tabacum* [13], *Oryza sativa* [14], *Vitis vinifera* [15], *Manihot esculenta* Grantz [16], *Camellia sinensis* [17], and the soybean [18]. 

The plant HXKs are classified into two major classes (type A and type B) and one minor class based on the amino acid sequences found in their N-termini (type C). Type-A HXKs contain chloroplast transit peptides and are located inside plastids, namely OsHXK4 in *O. sativa*, AtHXK3 in *A. thaliana*, and LeHXK4 in *L. esculentum*. Type-B HXKs, such as AtHXK1-2 and AtHLK1-3 (*A. thaliana*); OsHXK2, -3, -5, -6, -9, and 10 (*O. sativa*); and LeHXK1-3 (*L. esculentum*), share a common hydrophobic membrane anchor domain and are linked with mitochondria [19]. The *Physcomitrella patens* and *O. sativa* are the only ones with type-C HXKs, which do not have the anchor domain membrane or the plastidic transit peptide [20,21]. Due to their multiple subcellular localizations, HXKs are involved in many physiological processes. At least six members of the HXK family have been discovered in *Arabidopsis thaliana*; three of these members (AtHXK1, AtHXK2, and AtHXK3) encode active proteins with diverse cellular/tissue localization, while the remaining three lack catalytic function and are collectively known as HXK-like (HXL) protein [6]. Glucose-insensitive2 (gin2), a deletion mutant of AtHXK1, has been shown to function as a glucose sensor in sugar signaling [2,22,23]. *AtHXK1*, in conjunction with VHA-B1 and RPT5B, inhibits the transcription of its target genes by regulating the glucose signaling complex [24]. In addition to regulating growth and development, the *AtHXK1* gene family has been shown to have additional functions in several plant species [25,26]. The overexpression of *AtHXK1* in citrus resulted in an impairment of stomatal closure and acceleration of senescence. Overexpressing *AtHXK1* in transgenic tomatoes does not cause premature senescence under typical growing conditions [27]. It has been hypothesized in recent studies that *OsHXK5* and *OsHXK6* may also serve as glucose sensors [28]. Transgenic rice plants overexpressing these genes showed improved glucose insensitivity in *gin2-1* mutants and repressed the photosynthetic gene RbcS. This supplemented the glucose sensitivity of *gin2-1* that was provided by either *OsHXK5* or *OsHXK6*.

In addition, a significant amount of research has been carried out in regard to determining the plant HXK gene family’s important signaling function in response to a wide variety of environmental stresses [29]. The HXK pathway is involved in metabolic reprogramming in response to a variety of stresses, including phosphate starvation, cold, osmotic, and salt stress [30]. Recently, the *GmHXK2* gene was first isolated from the soybean, and overexpression of *GmHXK2* remarkably enhanced the tolerance of *Arabidopsis* transgenic plants to salt stress by maintaining the sodium (Na^+^) and potassium (K^+^) homeostasis, indicating the important role of HXK in regulating ion transport under salt stress [18]. HXK activities are also critical for maintaining normal levels of reactive oxygen species (ROS) in the face of stress ROS levels [31]. Ascorbic acid (ASA) biosynthesis is aided by HXK, which can have far-reaching effects on cytoplasmic reactive oxygen species (ROS) detoxification processes, cell elongation, and cell division [32], as ASA removes hydrogen peroxide via the Halliwell–Asada pathway. Ascorbic acid also aids in the regeneration of tocopherol, which adds an extra layer of protection to membranes.

The discovery of various HXK proteins in different plant varieties and the use of transgenic technology for the functional analysis of plant HXK genes have showed the importance of HXKs in the cell. However, there has been no comprehensive evaluation of the HXK gene family in the soybean genome. Therefore, detailed characterization will improve our understanding of their structure and function in this crop and help in crop improvement. The soybean, a major oil crop worldwide, is used for both human and animal nutrition and as a biofuel [33]. Soybean plant development, yield, and quality are all negatively impacted by alkali damage, which has become a global issue. In this study, we performed a genome-wide analysis to identify HXK-encoding genes in the soybean genome. The sequence analysis revealed 17 candidate HXK genes in the soybean genome. GmHXK protein promoters, exon–intron structures, synteny analyses, and evolutionary connections were all analyzed. By using qPCR (quantitative polymerase chain reaction), we were able to examine the temporal and spatial expression of *GmHXK1-17* in various organs. The differential expression and enzyme activity of *GmHXK1-17* were also analyzed under different biotic stress conditions. *GmHXK15* may play a role in the soybean alkali stress response, as its transcriptional response to alkali stress correlates with the activity of the HXK enzyme. To prove its expression in prokaryotes, we found that *GmHXK15* encodes an active HXK enzyme in *E. coli*. *GmHXK1-17* has been implicated in plant growth and stress responses, and this study is the first to report on the family’s complete identification and expression analysis in soybean.

## 2. Materials and Methods

### 2.1. Plant Materials

Soybean (*Glycine max* L.) plants of the SN14 variety were used throughout the study. (They were supplied by the Heilongjiang Academy of Agricultural Sciences in Harbin, China.) Plants were grown in a growth chamber with the diurnal temperature set to 23 ± 2 °C/20 ± 2 °C, light for 16 h daily, light intensity of 1000 µmol·m^−2^ s^−1^, and relative humidity of 60–80%. For the analysis of differential gene expression during seed development, the seeds were harvested at three different stages of development, as defined by the Soybase website (https://www.soybase.org/ontology.php, accessed on 7 September 2021): early-seed maturity stage (EM, SOY: 0001291), mid-seed maturity stage (MM, SOY: 0001293), and late-seed maturity stage (LM, SOY:0001293). To assess *GmHXK* transcriptional profiling under various levels of abiotic stress, soybean seedlings of the V2 (second trifoliate stage, SOY:0000017) stage were subjected to salt stress induced by 120 mM NaCl, alkali stress induced by 100 mM NaHCO_3_, and osmotic stress induced by 200 mM mannitol solutions. Roots were then collected at 0, 3, 6, 12, and 24 h after treatment and stored at −80 °C, with three biological replications, with untreated plants serving as controls.

### 2.2. Soybean HXK Gene Identification

To obtain all of the HXKs in the soybean genome, we used the previously published *A. thaliana* HXKs as queries in a BLASTP search against the soybean genetics and genomics database (SoyBase, http://www.soybase.org/sbt, accessed on 7 September 2021). Putative protein sequences belonging to members of the soybean HXK family that met the criteria of a sequence identity threshold of >90% and an E-value of less than 10^−10^ were included in the study. The SoyBase database contains information on the genetics of GmHXKs, such as the length of coding sequences, the location of genes on chromosomes, and the size of the proteins they encode. Calculations for molecular mass and isoelectric point were analyzed by using ExPASy’s ProtParam tool (https://web.expasy.org/protparam/, accessed on 12 September 2021) with default parameters.

### 2.3. Analyses of Evolutionary, Gene Structure, and Synteny of GmHXKs

The gene structure layouts were created using the Gene Structure Display Server (http://gsds.cbi.pku.edu.cn/index.php, accessed on 13 September 2021). The subcellular localization of proteins was identified by TargetP 1.1 (http://www.cbs.dtu.dk/services/TargetP/, accessed on 13 September 2021) and TopPred2 (http://www.sbc.su.se/erikw/toppred2/, accessed on 13 September 2021) with default parameters. A phylogenetic analysis was carried out by using the MEGA software with the bootstrap values performed on 1000 replicates to align HXKs from *G. max* (GmHXKs), *Z. mays* (ZmHXKs), *O. sativa* (OsHXKs), Phaseolus vulgaris (PvHXKs), *Medicago truncatula* (MtHXKs), *Sorghum bicolor* (SbHXKs), and *Brachypodium distachyon* [34]. The Plant Genome Duplication Database (PGDD, https://chibba.agtec.uga.edu/duplication/, accessed on 14 September 2021) was used to identify syntenic blocks among *Z. mays*, *G. max*, *A. thaliana*, *P. vulgaris*, *O. sativa*, *M. truncatula*, *S. bicolor*, and *B. distachyon* HXKs with default parameters [35]. HXK gene identifiers and other relevant data are listed in Appendix A.

### 2.4. Analysis of the Promoters of GmHXKs 

The SoyBase database (https://www.soybase.org/sbt, accessed on 14 September 2021) was used to identify the crucial cis-acting elements in the identified genes. The PlantCARE database (http://bioinformatics.psb.ugent.be/webtools/plantcare/html, accessed on 14 September 2021) was used to perform a *cis*-element scan on the 2 kb promoter region in the DNA sequence of each *GmHXKs* [36].

### 2.5. Expression Analysis of GmHXKs

High-throughput sequencing data from the Phytozome database were used to analyze the transcriptional profiles of *GmHXKs* across various plant organs and structures, including leaves, root hairs, root, stem, and flower tissues. The total RNA from root samples at 0 h, 3 h, 6 h, 12 h, and 24 h after treatment was isolated using a Quick Total RNA Isolation Kit (HUAYUEYANG, Beijing, China). The RNA samples were tested for concentration and quality and examined for structural integrity, using 1% agarose gel electrophoresis. Prime Script^®^ RT Master Mix (TakaRa Dalian, China) was used to create first-strand cDNA. The resulting cDNA was diluted to a concentration of 500 ng/L. For real-time quantification RT-PCR, the manufacturer’s protocols for SYBR Green (TaKaRa) were used, and PCR amplification was performed on a Roche LightCycler^®^ 96 q-PCR system. *GmHXK* transcripts in soybean roots grown under normal environmental conditions were utilized as a calibration standard. The *GmGAPDH* and *GmACTIN* genes were used as internal references. The qRT-PCR reaction included three biological replicates. The relative expression of target genes (*GmHXKs*) was calculated by using the ∆∆CT method: ΔΔCT = (CT(target, untreated) − CT(ref, untreated)) − (CT(target, treated) − CT(ref, treated)); fold-change  =  2^−△△ct^. The qRT-PCR primers are listed in Appendix A.

### 2.6. HXK Activity Assays

The HXK activity under various abiotic stress conditions was measured using an HXK assay kit from Solarbio Science and Technology (Beijing, China) according to the manufacturer’s protocol. HXK activity was measured in U/g fresh weight as total glucose-phosphorylating capacity. There were three independent technical analyses performed on each biological sample.

### 2.7. Expression of Recombinant Proteins and Determination of Enzyme Kinetic Properties

We used *NcoI* and *XhoI* restriction sites in the *GmHXK15* CDS to insert them into the pET32a (+) bacterial expression vector (for the vector map, see Appendix A). The putative recombinants were created by transforming *E. coli* (Rosetta) with the recombinant plasmid pET32a-GmHXK15. The expression of *GmHXK15* was induced by adding 1 mM of IPTG to liquid LB medium and incubating the cells at 37 °C for 4 h. The recombinant proteins were disrupted using 250 watts of ultrasonic waves for 10 min and then harvested using 10,000 g of centrifugal force for 15 min. Then, His-tagged proteins were purified by Ni-NTA affinity chromatography and detected by Western blot, using anti-his antibodies. The protein concentration was monitored by using a Bradford Protein Assay Kit purchased from Solarbio Science and Technology (Beijing, China). To obtain the K_M_ and V_max_ for substrates, progress curves were recorded using varying concentrations of glucose or fructose. The initial velocity was plotted against the substrate concentration and fitted with the Michaelis–Menten equation to obtain the kinetic parameters of GmHXK15 by GraphPad Prism version 8.0.1.

### 2.8. Overexpression of GmHXK15 in Soybean Hairy Roots

The complete coding regions corresponding to the putative mitochondria GmHXK15 gene were amplified from the roots of soybean cultivar “SN14” (for primers, see Appendix A). This gene was further constructed into the pBI121 vector, containing a green fluorescent protein (GFP) tag and a CaMV35S promoter (for the vector map, see Appendix A). The empty vector was used as the positive control. The resultant pBI121-GmHXK15::GFP fusion proteins were transiently transformed into the *Arabidopsis* mesophyll protoplasts, using the polyethylene glycol (PEG)-mediated protoplast transformation technique. In brief, 10 μg (about 20 μL) constructs were introduced into 100 μL of protoplast solution (~2 × 10^6^ cells). Then, 120 μL of PEG solution (40% PEG4000, 0.2 mol/L of mannitol, and of 0.1 mol/L CaCl_2_) was added to the tube, and transfection was initiated sequentially by a gentle tapping of the tube bottom 15 times. Transformed protoplasts were resuspended in 4 mL of liquid MS medium with 0.4 M sucrose and incubated at 24 °C in darkness, overnight. The localization of the expression of GFP was then visualized by confocal laser-scanning microscopy (LSM 710, Carl Zeiss, Jena, Germany) with respective excitation/emission wavelengths for GFP and chlorophyll autofluorescence of 488 nm/507–535 nm and 610 nm/650–750 nm. The recombination plasmids of pBI121-GmHXK15::GFP were independently transformed by electroporation into *Agrobacterium rhizogenes* strain K599, which was used to transform soybean hypocotyls. Soybean transformation in “DN50” hypocotyls and hairy root induction was performed as previously described [37]. Hairy roots from K599-infected soybean plants served as a control group. To examine the effects of transgenic hairy roots on many biochemical and physiological variables, more than 10–20 different transgenic hairy roots were investigated. The effects of 0 and 100 mM NaHCO_3_ on the maximum root length and fresh root weight of transgenic soybean hairy roots after 5 days of salt stress were examined.

### 2.9. Statistical Analysis

A minimum of three biological replicates were used per experiment. Results are means ± SD and were compared with Student’s *t*-tests, using SPSS 22.0. A *p* < 0.05 was the significance threshold.

## 3. Results

### 3.1. Detection and Characterization of Soybean’s Hexokinase Genes 

To find soybean HXK genes, *Arabidopsis*-derived HXK protein sequences, including AtHXK1 (AT4G29130.1), AtHXK2 (AT2G19860.1), AtHXK3 (AT1G47840.1), AtHXL1 (AT1G50460.1), AtHXL2 (AT3G20040.1), and AtHXL2 (AT4G37840.1), were used as query sequences in a search out of the soybean genome. Following the investigation of conserved domains, three sequences were eventually eliminated from the analysis of 17 candidate sequences, namely GmHXK1-GmHXK17 (Appendix A). The molecular weight (MW) of AtHXK or GmHXK proteins varies widely, as does their CDS length. The length of AtHXK and GmHXK proteins ranged from 280 amino acids (aa) (AtHXK1) and 156 aa (GmHXK1) to 503 aa (AtHXK1) and 504 aa (GmHXK9 and GmHXK10), respectively. The molecular weights of AtHXK and GmHXK proteins ranged from 31.07 kDa (AtHXK1) and 17.42 kDa (GmHXK1) to 54.96 kDa (AtHXL2) and 55.04 kDa (GmHXK9), respectively. The protein isoelectric point of AtHXK and GmHXK proteins ranged from 5.55 (AtHXL1) and 5.11 (AtHXL2) to 8.90 (GmHXK4) and 8.76 (GmHXK7), respectively. Appendix A provides further details about the individual GmHXKs, such as their pI and gene location. In the structural study of translated proteins for GmHXK2-17, the bi-domain structure of GmHXK proteins was found to be comparable to that of *A. thaliana* HXK proteins, with an N-terminal Hexokinase 1 domain (PF00349) and a C-terminal Hexokinase 2 domain (PF03727), while GmHXK1 protein have only a C-terminal Hexokinase 2 domain (PF03727) (Figure 1).

### 3.2. Phylogenetic Relationships and Multiple Alignments

To examine the evolutionary relationships between GmHXK and other plant HXKs, a phylogenetic tree was constructed using 51 full-length HXK sequences. This included six HXKs from *A. thaliana* and ten from *Oryza sativa*, five protein sequences from Medicago truncatula, seven protein sequences from *Phaseolus vulgaris*, seven protein sequences from *Sorghum bicolor*, and sixteen protein sequences from *G. max*. Chromosomal localization sequences and associated accession numbers can be found in Appendix A. In Figure 2A, GmHXK members were assigned to one of three evolutionary clusters (I, II, and III). GmHXK1-4 were found in Cluster I, together with two other type-A HXKs (AtHXK3 and OsHXK4) that contain the chloroplast transit peptide. Clade II comprised type-B isoforms, containing GmHXK5-17, as well as eleven type-B HXKs (OsHXK2, -3, -5, -6, -9, and -10; AtHXK1-2; and AtHLK1-3). Three OsHXKs (OsHXK1, -7, and -8) were identified as type-C isoforms; these proteins lack membrane-anchored domains and chloroplast transport peptides, placing them in Clade III. No similar sequences of type-C OsHXKs were found in the phylogenetic tree of soybean plants. The in silico prediction made by GmHXKs indicated that the phylogenetic clades were comparable to one another.

The protein sequences were aligned using the Seaview 4 software for further characterization of the GmHXK proteins (Figure 2B). Previous *Arabidopsis* HXK studies were utilized to anticipate conserved sequences. Most GmHXK proteins have three conserved domains: an adenosine-phosphate-binding, a substrate-binding, and an ATP-binding domain with P1 (phosphate 1) and P2 (phosphate 2) residues (phosphate 2). Unlike GmHXK11, GmHXK2 lacks the N-terminal sequence and possesses the substrate-binding and the conserved P1 (in the ATP-binding domain) domain (Figure 2B). Four type-A isoforms (GmHXK1-4) and thirteen type-B isoforms of GmHXKs were identified by phylogenetic and protein structure analyses (GmHXK5-17). The localization feature of soybean HXK genes reveals the evolutionary relationship between the distinct function of HXK isoforms in each clade.

### 3.3. Syntenic Relationships between GmHXK Genes and Their Gene Structures

A synteny analysis among *HXKs* from *G. max*, *P. vulgaris*, *O. sativa*, *A. thaliana*, *M. truncatula*, and *S. bicolor* was performed in the present study to gain some insight into the potential function of *GmHXKs*. As shown in Figure 3A, the seventeen *GmHXKs* were scattered along ten out of twenty soybean chromosomes, and each of the ten chromosomes comprised one-to-three *GmHXKs*. Moreover, a total of 30 orthologous pairs of *HXKs* were found in the above six species (Figure 3A and Appendix A). The *GmHXKs* had a syntenic relationship only with *PvHXKs*, *AtHXKs*, *MtHXKs*, and *SbHXKs*, including 3 orthologous gene pairs between *G. max* and *S. bicolor*, 11 orthologous gene pairs between *G. max* and *A. thaliana*, 16 orthologous gene pairs between *G. max* and *M. truncatula*, and 22 orthologous gene pairs between *G. max* and *P. vulgaris* (Figure 3A). In addition, 24 paralogous *HXK* gene pairs were confirmed in the soybean genome, and the paralogous gene pairs were apt to be found among the members in the same subfamily (Figure 3A). These synteny occurrences proved that many MDH genes had already evolved before the divergence of soybean species. Genome and coding sequence (CDS) data for *GmHXKs* are used by the GSDS tool to generate an exon–intron prediction. A majority of *GmHXK* genes have nine exons and eight introns, whereas *GmHXK5* and *GmHXK11* have ten exons and nine introns. Based on these findings, the gene sequences and exon–intron organization of the GmHXKs genes were highly conserved.

### 3.4. The GmHXKs Promoters Possess Regulatory Elements

The transcriptional regulation of *GmHXKs* was investigated by examining a 2.0 kb upstream promoter region that precedes the ATG translation start codon. As can be seen in Figure 4, most of the *GmHXK* genes contain a small number of stress-responsive cis elements. For example, the MBS cis element, which is involved in drought responsiveness, was discovered in *GmHXK1-5*, *GmHXK8*, *GmHXK9*, *GmHXK11-13*, and *GmHXK15*. A cis element that associates defense and stress responsiveness (TC-rich repeats) with each other was found in the *GmHXK17* and *GmHXK13* genes; LTR, a cis element involved in low-temperature response was found in the *GmHXK5*, *GmHXK8*, *GmHXK15*, *GmHXK3-4*, *GmHXK7*, *GmHXK9*, and *GmHXK10-11* genes (Figure 4). All *GmHXK* genes have cis elements in their promoters that are involved in at least one of several different hormone response pathways. These include the anaerobic response element (ARE), the ethylene response element (ERE), the salicylic acid response element (SRE), the auxin response element (ABRE), and the MeJA response element (TGAGG and CGTCA).

### 3.5. The Expression Profiles of GmHXK during Different Stages of Developmental and Tissues

An analysis of the *GmHXK* transcriptional patterns in leaves, roots, root hairs, shoot apical meristems, nodules, stems, seeds, pods, and flowers was performed using Phytozome high-throughput sequencing data. It was found that, among all *GmHXK* genes, *GmHXK3*, *12*, *15*, and *16* were expressed most strongly in all of the tissues examined (Figure 5A). The highest levels of expression of *GmHXK3* were found in pods, followed by root hairs, roots, and flowers. Three genes (*GmHXK17*, *GmHXK13*, and *GmHXK8*) showed a high expression in the root system, while three genes showed a low expression in the root system (*GmHXK4*, *GmHXK9*, and *GmHXK10*). The expression of *GmHXK11* was primarily localized in seeds and nodules, while *GmHXK5* was only expressed in flowers. All of the tested tissues and organs expressed low levels of *GmHXK1*, *GmHXK2*, *GmHXK6*, and *GmHXK14* (Figure 5A). The versatility of *GmHXKs* in soybean growth and development is reflected in these tissue-specific expression patterns.

The expression of the *GmHXK* gene in maturing soybean seeds was analyzed using quantitative real-time polymerase chain reaction (qRT-PCR) at 14, 25, and 35 days after flowering (40 DAF). Using the expression of *GmHXK14* in developing seeds, researchers compared it to the expression of other genes (Figure 5B). When compared to the expression of other *GmHXK* genes, *GmHXK3* and *GmHXK16* were consistently more highly expressed across all developmental stages, with peak expression occurring at the MM and LM stages, respectively. Throughout, *GmHXK14* expression remained at low levels. The *GmHXK13*, -*14*, -*15*, and -*7* genes were most highly expressed during the LM stage of growth, while the *GmHXK4*, *GmHXK12*, and *GmHXK8–10* genes were most highly expressed during MM stage growth (Figure 5B).

### 3.6. Abiotic-Stress-Induced Enzyme Activity and Transcript Level of GmHXKs

The significance of *HXK* genes in stress adaptation has been shown in a variety of model plants. Furthermore, the promoter analysis of soybean HXKs revealed transcription binding motifs and cis elements (MBS, LTR, and TCA elements) that are likely to contribute to the response to salt and drought stress, which are common in our region (Northeast China). We tested the effects of alkali (100 mM NaHCO_3_), salt (120 mM NaCl), and osmotic (200 mM mannitol) on *GmHXKs* to see how they react to stress. Upon exposure to abiotic stress, *GmHXKs* showed dramatically different transcriptional profiles (Figure 6). In response to salt treatment, transcript levels for many *GmHXKs* rose, with *GmHXK7* showing the most dramatic increase at 12 h. In addition, transcription levels in *GmHXKs* genes were generally highest after 6 or 12 h of salt treatment, with *GmHXK16* showing the highest levels. When compared to other treatments, alkali treatment resulted in significantly higher *GmHXKs* transcript levels (Figure 6). Most *GmHXK* genes were activated in response to alkali treatment. While the majority of genes showed significant upregulation after 12 h, Gm*HXK6*, -*8*, and -*13* were induced after 6 h and kept relatively high transcript levels during the treatment. After being exposed to alkali for 12 h, *GmHXK15* transcripts were significantly higher than those of other genes. The transcripts of *GmHXK2*, -*6*, and -*8* after 6 h exhibited a marked upregulation. Similarly, most *GmHXKs* were stimulated after 12 h of mannitol treatment, maintaining a relatively high-abundance transcript level throughout the entire treatment period.

The enzymatic assay showed that the activity of soybean HXK was significantly raised after treatment with NaCl, NaHCO_3_, and mannitol (Figure 7A). The above treatments increased the HXK activity by a factor of ten to twenty. The alkali treatment stimulated HXK activity more than the other treatments, leading to a rapid increase in HXK activity within 12 h. (Figure 7A). After 12 h, HXK activity following salt and osmotic treatment reached its peak before gradually declining. An analysis of the correlations showed that the HXK activity under abiotic stress conditions was consistent with the transcripts of *GmHXK15*, suggesting that *GmHXK15* encodes the primary HXK isoform involved in responses to abiotic stress (Figure 7B).

### 3.7. GmHXK15 Prokaryotic Expression and Subcellular Localization Analysis

The recombinant protein was isolated by expressing His-tagged *GmHXK15* in *E. coli* DE3 cells, and the catalytic properties of the GmHXK15 gene were investigated further to understand its resistance functions. SDS-PAGE and Western blotting were used to purify the protein to a single band with the expected molecular mass (Figure 8A). The kinetic parameters of GmHXK15 in relation to glucose and fructose were calculated using Eadie–Hofstee data plots. It was found that the Vmax and Km values for glucose were 0.27 mol·min^−1^·mg^−1^ of protein and 0.06 mM, respectively (Figure 8B), and that the corresponding values for fructose were 0.71 mol·min^−1^·mg^−1^ of protein and 1.3 mM. (Figure 8C). As a result, *GmHXK15* had a higher affinity for glucose than fructose. Although this is the case, fructose has greater maximal activity than glucose. These findings suggest that GmHXK15 participates in hexose phosphorylation.

To further certify the subcellular localization of *GmHXK15*, the entire coding regions were successfully cloned and verified. The transient expression of GFP-tagged GmHXK15 and positive control (35S: GFP) proteins was performed in *Arabidopsis* mesophyll protoplasts. However, whereas *GmHXK15* was selectively localized to the mitochondria, free GFP was found in all cellular compartments, except for the chloroplast and vacuole (Figure 8D). These findings agreed with the prior online prediction and may have larger ramifications for comprehending the subcellular localization of plant HXKs.

Next, soybean hairy roots with overexpressed *GmHXK15* were created, and *GmHXK15* transcripts were found to be 1.3–6.1-fold higher than in control hairy roots (CHRs; Appendix A). We selected two OHR lines (OHR1 and OHR2) with high expression levels of *GmHXK15* for additional investigations. The overexpression of *GmHXK15* had little influence on soybean growth under normal conditions (Figure 8D). However, after exposure to NaHCO_3_ treatment, the OHR plants exhibited greater alkali resistance, as seen by increased root elongation and fresh root weight, compared to the CHR plants (Figure 8E,F). These findings on the effects of overexpression supported the hypothesis that chloroplast *GmHXK15* positively controlled alkali tolerance in soybeans.

## 4. Discussion

Hexose sugar, the main carbon source of energy storage, is essential for cellular life and metabolism [38]. Hexoses can also function as signaling molecules, playing roles in a wide range of cellular processes [2]. To be used, hexoses must first be phosphorylated, a process that can be carried out by hexose-phosphorylating enzymes such as HXKs [2,22]. Hexokinases (HXKs) have been discovered in a wide variety of plant species due to the wide range of functions served by hexokinases that phosphorylate fructose and glucose. So far, few details about the function of the *HXK* gene in soybeans have been reported. Seventeen *HXK* genes (*GmHXK1-17*) were mapped out in the soybean genome in this study (Appendix A), which was similar to that reported in a recent study [18]. The phylogenetic analysis indicated that *GmHXK1*, -*2*, -*3*, and -*4* belonged to the type-A subfamily group, while *GmHXK5-17* were in the type-B subfamily group (Figure 1, Figure 2 and Figure 3). All seventeen *GmHXK* genes were found on ten different chromosomes (Figure 3). A great deal of variation has been found within the *HXK* gene family in plants; for example, *A. thaliana* has six subfamilies [22], *Zea mays* has nine [12] and *Oryza sativa* has ten [14], all of which are dispersed across various chromosomes. All *GmHXKs*, except for *GmHXK5* and *GmHXK11,* are encoded by a total of nine exons (Figure 3). There are clear similarities between the intron–exon structure of the *GmHXKs* and that of other identified plant *HXK* genes.

Herein, the phylogenetic analysis of the 51 HXK proteins from six plants showed that *GmHXKs* displayed a particular similarity for HXKs from *P. vulgaris* and *M. truncatula* of leguminous species (Figure 2). Furthermore, all the *GmHXKs* were found to be closely related to their genes corresponding in common bean and lucerne, which is consistent with their genetic associations, implying that they may be functionally conserved. Our results also show that *GmHXKs* diverged before the divergence of sorghum, rice, common bean, and lucerne, suggesting that *HXKs* evolved before these four major plant groups diverged (Figure 3). For most, a high degree of sequence similarity is associated with conserved functions [3].

The well-characterized tertiary structures of yeast and mammalian HXK proteins reveal that they are composed of a large and a small domain, the core of which is primarily composed of sugars [39,40]. This indicates that a majority of the conserved amino acid residues at the interface between the two domains are responsible for generating the glucose and ATP-binding sites [40,41]. Here, in this study, multiple sequence alignments of HXK homologous soybean proteins highlighted complete conservations that are consistent with the reported HXKs, including motifs with names such as phosphate 1, phosphate 2, adenosine, and sugar-binding (Figure 2). *GmHXK1* and *GmHXK2* exhibited some observable deviations. Phosphate 1 (P1) and a substrate-binding domain are both absent in *GmHXK1* and *GmHXK2*. In addition to this, *GmHXK1* and *GmHXK2* also have a deletion in the P2 motif. Because of the high conservation of the P1 domain and the requirement for an HXK activity [40], these proteins have almost certainly lacked HXK activity. However, a recent study analyzed the molecular characteristics of *GmHXK2* and found that *GmHXK2* gene played an important role in resisting salt stress [18].

Understanding plant growth and development requires an understanding of how proteins are produced and localized in various tissues. Unlike *AtHKL3*, which is exclusively found in the flower, *Arabidopsis* expresses the majority of its *HXK* genes in all tissues [22,25]. Genes in the *OsHXK* family of rice all display comparable expression profiles, except for *OsHXK10*, which is only active in bloom, and OsHXK1, which cannot be detected [20]. With the exception of *AtHKL3*, *OsHXK1*, and *OsHXK10*, all of the *Arabidopsis* and rice *HXK* genes were detectable in key tissues, suggesting that they may each serve a unique or redundant purpose in these diverse organisms. Leaves, roots, root hairs, shoot meristems, nodules, stems, seeds, pods, and flowers were all analyzed for *PeHXK* gene expression patterns here in this study [42]. The expression patterns of the genes encoding *GmHXKs* were like those observed in other plants, such as rice and *Arabidopsis.* Like the *AtHXK* and *OsHXK* genes, most *GmHXK* family members are substantially expressed in the root. *OsHXK10*, *GmHXK5*, and *GmHXK6* were barely detectable in any of the tissues examined. Similar to their rice homologs *OsHXK5*, *OsHXK6*, and *OsHXK9*, *GmHXK15*, *GmHXK17*, and *GmHXK7* were significantly more expressed in leaves and stems (Figure 5).

In addition to controlling plant growth, plant HXKs have also been shown to play a role in stress-response mechanisms [12,28,31]. Several *cis*-acting elements in *GmHXK* promoters may contribute to plant responses to drought and salt stresses, including MBS, ABRE, ARE, and TC-rich repeats (Figure 4). We also used qRT-PCR and enzyme activity assays to investigate the transcriptional and physiological responses to salt, alkaline, and osmotic stress, and we observed significant expression alterations in *GmHXKs* under these conditions (Figure 6 and Figure 7). *A. thaliana*, *Nicotiana benthamiana*, and the *Camellia sinensis* (tea plant) are just a few examples of plants that have been shown to respond to abiotic stress via response multiple *HXKs* [6,17,22,27]. Most *HXKs* showed strong expression increases in response to environmental stresses such as cold, salt, drought, and heat [4,42]. *CsHXKs*’ expression was reported to increase in response to cold, salt, drought, and exogenous ABA treatments [17]. Another study found that *NbHXK1* played a role in the organism’s reaction to oxidative stress brought on by methyl viologen. Under abiotic stress, Figure 7 shows a significant correlation between the expression patterns of the type-B isoform (*GmHXK15*) and the activity levels of the HXK enzyme, indicating a major role in the response. The same results were observed in a previous study: *GmHXK15* was clearly stimulated under drought and salt stresses, but the precise regulatory mechanism has not been thoroughly studied [18]. Two type-B isoforms, ZmHXK5-6, play a positive function in salt-induced responses in maize [12].

Previous research has shown that *AtHXK2*-encoded type-B isoforms are stimulated for action under osmotic and salt stress [30,31,43]. Based on these different expression patterns, it appears that *GmHXKs* are involved in regulating the physiological response of the soybean plant to abiotic stress. Importantly, a mitochondrial isoform (*GmHXK15*) was found to respond more rapidly and strongly to alkali treatment than other genes, achieving its maximal transcriptional induction within 12 h of alkali stress (Figure 6), as is consistent with the transcriptional profiles of its orthologous genes [28,32].

The root length and fresh root weight both noticeably increased when *GmHXK15* was overexpressed in transgenic soybean hair roots, thus indicating the plant’s increased alkali tolerance (Figure 8). Further research revealed that *GmHXK15* can catalyze the phosphorylation of hexose (Figure 8). Despite having a lower Km value for glucose and a higher V_max_ value for fructose, *GmHXK15* has a lower Km value for glucose. Compared to glucose, *MeHXK2* has a lower K_m_ value and a higher V_max_ value for fructose [16]. Based on these results, the two characteristics of plant *HXKs* were confirmed: their low substrate selectivity and their preference for glucose over fructose. These findings confirmed two characteristics of plant *HXKs*: their low substrate selectivity and preference for glucose over fructose. Previous research has shown that *AtHXK2*-encoded type-B isoforms are stimulated for action under osmotic and salt stress [30,31]. Based on these different expression patterns, it appears that *GmHXKs* are involved in regulating the physiological response of the soybean plant to abiotic stress. Further research revealed that *GmHXK15* can catalyze the phosphorylation of hexose (Figure 8). Despite having a lower Km value for glucose and a higher V_max_ value for fructose, *GmHXK15* has a lower Km value for glucose. Compared to glucose, *MeHXK2* has a lower K_m_ value and a higher V_max_ value for fructose [16]. These findings confirmed two characteristics of plant HXKs: their low substrate selectivity and preference for glucose over fructose.

## 5. Conclusions

So far, 17 *HXK* genes have been discovered in the soybean genome. Based on predictions of subcellular localization and phylogenetic analysis, *GmHXKs* were separated into type-A and -B isoforms. In response to abiotic stresses, the expression of *GmHXK* genes changes, reflecting their potential functional differences. The expression of the type-B HXK (*GmHXK15*) gene was positively correlated with HXK activity, suggesting its central role in response to alkali stress. A subsequent study confirmed that the *GmHXK15* gene encodes a functional HXK enzyme. The overexpression of *GmHXK15* in soybean hairy roots resulted in elevated tolerance to alkali stress. The current work lays the groundwork for HXKs, which will facilitate future cloning and analysis of *GmHXK* gene function in the soybean.

## Figures and Tables

**Figure 1 plants-12-03121-f001:**
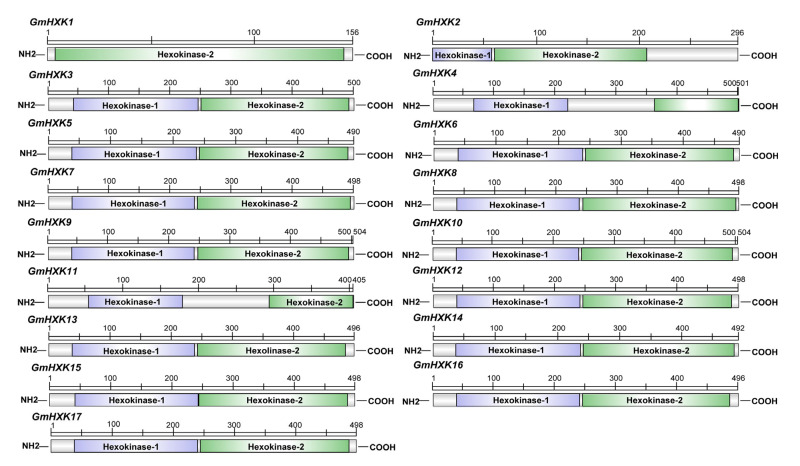
The domain structure found in the soybean HXK proteins.

**Figure 2 plants-12-03121-f002:**
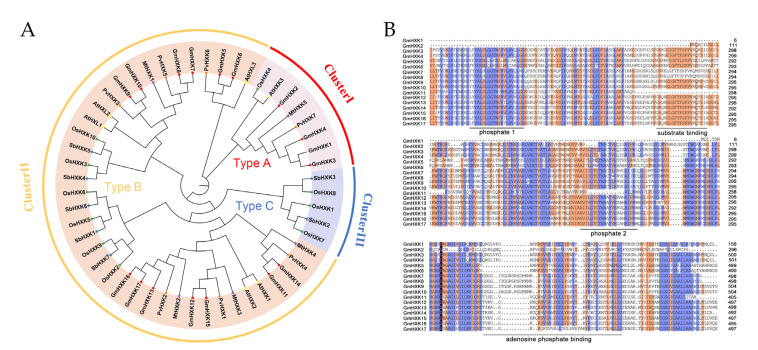
Phylogenetic and multiple alignments analysis of GmCKs. (**A**) Phylogenetic tree of HXK proteins from *G. max*, *A. thaliana*, *O. sativa*, *P. vulgaris*, *M. truncatula*, and *S. bicolor*. (**B**) Multiple alignments of protein sequences of soybean HXKs.

**Figure 3 plants-12-03121-f003:**
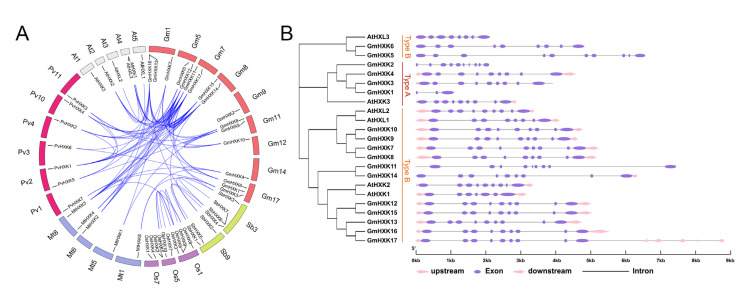
An analysis of the GmHXK family of genes’ synteny and intron–exon arrangement. (**A**) Syntenic assessment of soybean *HXKs* with respective genes from *A. thaliana*, *G. max*, *P. vulgaris*, *O. sativa*, *S. bicolor*, and *M. truncatula*. Chromosomes are represented by circles for the species above. HXK collinearity is indicated by colored curves. (**B**) The intron–exon organization of *AtHXKs* and *GmHXKs.* A pink arrow indicates an untranslated region (UTR). Alternatively, colored ellipses represent exons, while gray lines represent introns.

**Figure 4 plants-12-03121-f004:**
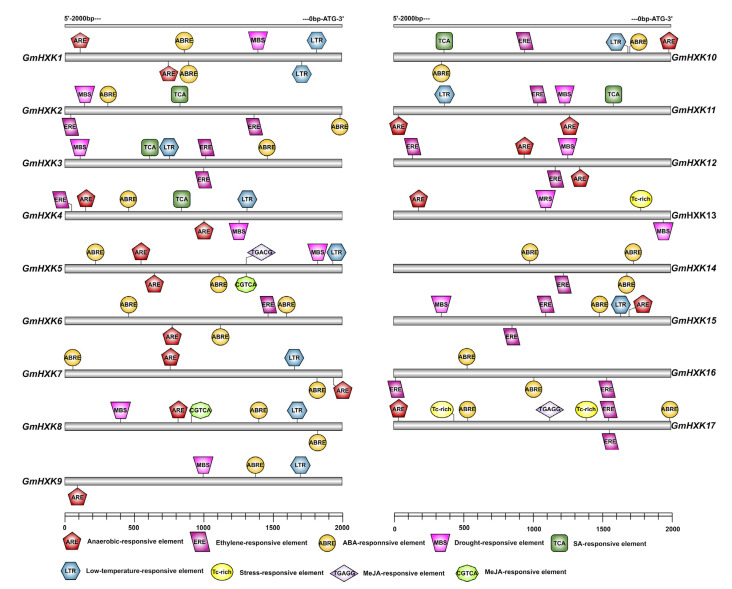
Upstream of the GmHXKs start codon, cis elements are predicted in the 2.0 kb promoter region. Genes encoding *GmHXK* are colored based on their respective positions of cis elements.

**Figure 5 plants-12-03121-f005:**
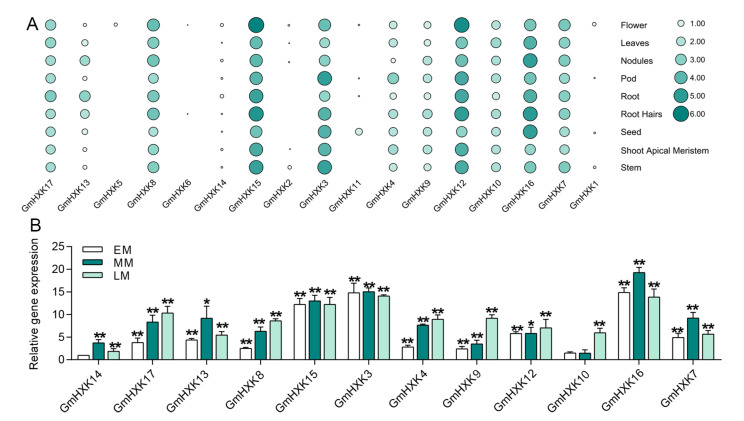
The *GmHXK* is expressed in a variety of tissues and developmental stages. (**A**) Cluster analysis of expression profiles of *GmHXKs* that participate in tissue development. The Phytozome database was used to examine transcripts of *GmHXK* genes in various tissues. Heat maps illustrate the results. A deep color denotes an elevated level of transcription, and a light color denotes low-level transcription; the color scale represents log2 expression values. An enlarged circle indicates a high level of transcription, while a smaller circle indicates low levels. (**B**) An assessment of transcript profiles of *GmHXKs* in developing seeds at early maturation (EM), mid-maturation (MM), and late maturation (LM). The transcripts of *GmHXK6* were used as internal references in developing seeds at the EM stage. Each tissue was analyzed in three biological replicates. Statistically significant differences from the control group, as determined by Student’s *t*-test, are indicated by asterisks above the bars (* *p* < 0.05; ** *p* < 0.01).

**Figure 6 plants-12-03121-f006:**
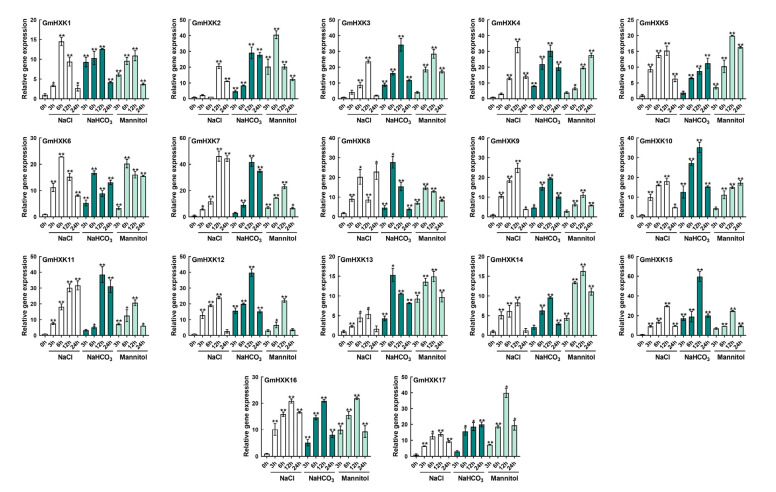
*GmHXKs* expression profiles in soybean plant roots subjected to 120 mM of NaCl, 100 mM of NaHCO_3_, 200 mM of mannitol, or water (control) for 0, 6, 12, and 24 h, each. *GmHXKs* were estimated under non-stressed conditions to function as a calibrator. Each experiment was performed in triplicate. Moreover, the qRT-PCR results were examined using the 2^−ΔΔct^ method. Asterisks above bars signify a difference regarding statistical significance, as determined by Student′s *t*-test (* *p* < 0.05, ** *p* < 0.01).

**Figure 7 plants-12-03121-f007:**
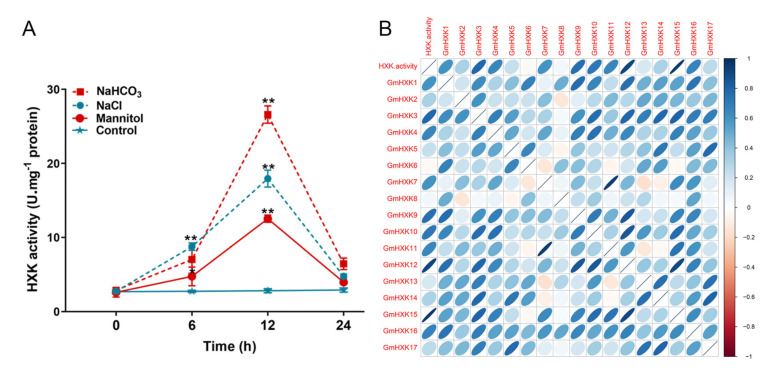
In soybean, HXK activity levels were determined under abiotic stress treatments. (**A**) HXK activity was evaluated for 0 h, 6 h, 12 h, and 24 h in roots of soybean plants treated with 120 mM of NaCl, 100 mM of NaHCO_3_, 200 mM of mannitol, or water (control). Biological replicates were represented by means ± SD. Statistically significant differences from the control group, as determined by Student’s *t*-test, are indicated by asterisks above the bars (* *p* < 0.05; ** *p* < 0.01). (**B**) The correlation coefficient between HXK enzymatic activity and the expression levels of GmHXKs. Correlation coefficients between any two traits are shown as ellipses on each chart. The correlation magnitude is determined by the color and slope of the ellipse. Negative correlations are represented by a red ellipse, and positive correlations by a blue ellipse.

**Figure 8 plants-12-03121-f008:**
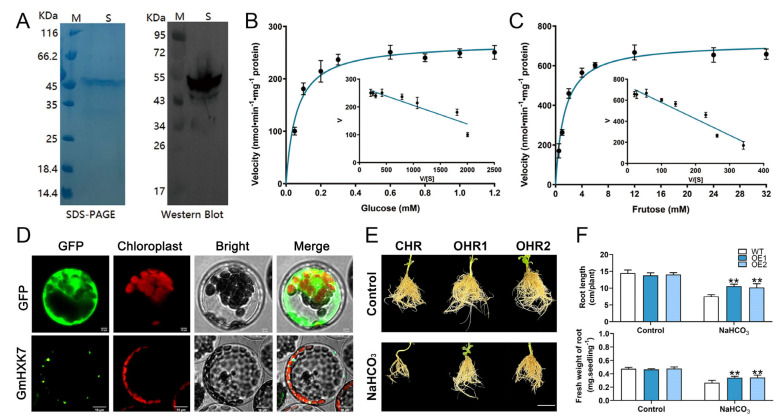
Overexpression of *GmHXK15* confers alkali tolerance in transgenic soybean hairy roots. (**A**) Purified GmHXK*15* fusion proteins were resolved by SDS-PAGE and Western blotting (Lane S). The Eadie–Hofstee plot depicts the kinetic characteristics of GmHXK*15* in response to (**B**) glucose and (**C**) fructose. (**D**) Subcellular localization analysis of GmHXK*15* by expressing GmHXK-GFP fusion protein transiently in *Arabidopsis* mesophyll protoplasts. (**E**) Performance and (**F**) maximum root lengths and root fresh weights of *GmHXK15* -OHR and CHR plants subjected to 100 mM NaHCO_3_ over 5 days. Bars = 5 cm. Asterisks denote significant differences (** *p* < 0.01) from CHR plants, as shown by Student’s *t*-test.

## Data Availability

Not applicable.

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
