# Peer review of "Genome-Wide Characterization of Soybean Hexokinase Genes Reveals a Positive Role of GmHXK15 in Alkali Stress Response"

_plants, 2023, doi:10.3390/plants12173121_

Round 1

Reviewer 1 Report

In this study, Jiao et al. reported the genome-wide identification and characterization of Hexokinase genes in soybean. In total 16 HXK genes were systemically analyzed. Authors found that some of the members play dominant roles in response to abiotic stimuli. Importantly, by using genetic approach, authors found that GmHXK7 was specifically targeted to the mitochondria, and overexpression of GmHXK7 gene protects soybean from alkali stress. Overall, this study is well designed and performed in a high standard. The conclusions are well supported by the bioinformatic and experimental data. Data are presented in a logic way with stringent statistical analysis. I have no further comments regarding this manuscript, and suggest to the next step during the editorial process.

NA

Author Response

Thank you very much for your time on reviewing our manuscript. We have carefully read the nice comments from you and found that these suggestions are very helpful for us to improve our manuscript. According to your comments, we carefully checked the whole manuscript, including spelling check, grammar check, etc, and made some changes to improve the manuscript. The changes were tracked and revised manuscript was resubmitted.

Reviewer 2 Report

This manuscript is an interesting investigation into the hexokinase gene family, specifically examining the family in Glycine max.  Through comparative genomics, the authors identified 16 HXKs in G. max and were able to classify them both through phylogenic analyses and through peptide domains to help predict localization.  The qRT-PCR demonstrating whether each GmHXK plays a role in different stresses and the baseline expression of each GmHXK in different tissues and at different timepoints was excellent.  However, work in soybean has shown it to have a very fast response to stresses.  By not looking at stress responses until 6 hours after stress exposure starts, you are missing the initial responses. These are downstream responses.  A statement should be added to address this limitation in the study.

While the manuscript is fairly well written, it does need some clarifications throughout as it’s unclear at some points what genes are being discussed (just the G. max HXKs, all known HXKs, other options).  The way it's currently written it's confusing what the 16, 22, and 51 genes are referring to (see comments below). 

Lines 58-62: Are OsHXK5 and OsHXK6 homologous to AtHXK1?

Ln 72: “as it removes hydrogen peroxide”, is the “it” Ascorbic acid or HXK?

Ln 124-126: A paragraph should be more than one sentence.  Please change sentence to read “the SoyBase database (https://www.soybase.org/sbt) was used to identify the crucial cis-acting elements in the identified genes.  The PlantCARE database (http://bioinformatics.psb.ugent.be/webtools/plantcare/html) was used to locate the cis-acting regulatory DNA elements. 

Ln 131 – 135: Were GmHXKs in roots or soybean leaves used as the calibration standard? 

Ln 197: Phaseolus vulgaris should be italicized. Same with G. max on Ln 198

Ln 232: To make it easier on the reader, please list the PvHXKs, OsHXKs, AtHXKs, SbHXKs, and MtHXKs in the same order as they’re explained in the second half of the sentence (SbHXKs, AtHXKs, MtHXKs). 

Ln235: Do the gene pairs described in the first part of the sentence represent 22 orthologous genes?  Line 81 states there are only 16 GmHXKs, so that statement needs more words to be interpreted by the reader.

Ln236: How are there 24 paralagous HXKs in the soybean genome if there are only 16 (ln 81) or 22 (Ln235).  Need clarifications.

Ln 394-395: Remove () from GmHXK1-7, 10, and 12-16.

Ln402: Now there are 51 GmHKXs?

I like the discussion, well written and reasoned. 

Author Response

Reviewer #2:

This manuscript is an interesting investigation into the hexokinase gene family, specifically examining the family in Glycine max. Through comparative genomics, the authors identified 16 HXKs in G. max and were able to classify them both through phylogenic analyses and through peptide domains to help predict localization.  The qRT-PCR demonstrating whether each GmHXK plays a role in different stresses and the baseline expression of each GmHXK in different tissues and at different timepoints was excellent.  However, work in soybean has shown it to have a very fast response to stresses. By not looking at stress responses until 6 hours after stress exposure starts, you are missing the initial responses. These are downstream responses.  A statement should be added to address this limitation in the study.

While the manuscript is fairly well written, it does need some clarifications throughout as it’s unclear at some points what genes are being discussed (just the G. max HXKs, all known HXKs, other options). The way it's currently written it's confusing what the 16, 22, and 51 genes are referring to (see comments below). 

Response: Dear reviewer, thank you very much for your help on revising our manuscript, and pointing out our problems. We have made correction according to your comments, We have supplemented the initial responses of GmHXKs after stress exposure starts (the newly submitted Fig 6), and the revised manuscript has been submitted online. We also carefully checked the whole manuscript, including spelling check, grammar check, etc, and made some changes to improve the manuscript. The detailed responses to your comments are listed in this online review forum.

Major comments:

1.Lines 58-62: Are OsHXK5 and OsHXK6 homologous to AtHXK1? Ln 72: “as it removes hydrogen peroxide”, is the “it” Ascorbic acid or HXK?

Response: Thank you very much for your time on reviewing our manuscript. 1)OsHXK5 and OsHXK6 are homologous to AtHXK1. 2)The HXK-derived glucose-6-phosphate can feed the l-galactose route in the so-called Smirnoff-Wheeler pathway, leading to biosynthesis of ascorbic acid (ASA), which has major implications in the well-known cytoplasmic ROS detoxification processes, cell elongation and, possibly, and cell division. ASA works in close co-operation with glutathione to remove hydrogen peroxide via the Halliwell-Asada pathway. In the sentence “as it removes hydrogen peroxide”, “it “ represents ASA. Sorry about this unclarity, we have revised some sentences or descriptions in the newly submitted manuscript to make it more accurate. The changes were highlighted in yellow (Introduction section, lines 57-64, lines 70-73).

2.Ln 124-126: A paragraph should be more than one sentence.  Please change sentence to read “the SoyBase database (https://www.soybase.org/sbt) was used to identify the crucial cis-acting elements in the identified genes.  The PlantCARE database (http://bioinformatics.psb.ugent.be/webtools/plantcare/html) was used to locate the cis-acting regulatory DNA elements. 

Response: Thank you very much for your help on revising these sentences, and pointing out our problems. The changes were highlighted in “Materials and Methods" part of the revised manuscript (Materials and Methods section, lines 136-140).

3.Ln 131 – 135: Were GmHXKs in roots or soybean leaves used as the calibration standard? Ln 197: Phaseolus vulgaris should be italicized. Same with G. max on Ln 198.Ln 232: To make it easier on the reader, please list the PvHXKs, OsHXKs, AtHXKs, SbHXKs, and MtHXKs in the same order as they’re explained in the second half of the sentence (SbHXKs, AtHXKs, MtHXKs). 

Response: Thank you for pointing out our problems. 1) GmHXK transcripts in soybean roots grown under normal environmental conditions were utilized as a calibration standard. We are very sorry for this negligence, and the word “leaves” has been corrected to “roots” (Materials and Methods section, lines 151-153). 2) we thank the reviewer for pointing out this problem, and “Phaseolus vulgaris” and “G. max” has been corrected to “Phaseolus vulgaris” and “G. max” in the new manuscript. According to your advice, we have revised the whole manuscript carefully and made sure to define the species name in italics (Results section, lines 243-244). 3) We thank the reviewer for reminding us this problem. As you advised, we have list the PvHXKs, OsHXKs, AtHXKs, SbHXKs, and MtHXKs in the same order. The changes were tracked and revised manuscript was resubmitted (Results section, lines 271-283).

4.Ln235: Do the gene pairs described in the first part of the sentence represent 22 orthologous genes?  Line 81 states there are only 16 GmHXKs, so that statement needs more words to be interpreted by the reader.Ln236: How are there 24 paralagous HXKs in the soybean genome if there are only 16 (ln 81) or 22 (Ln235).  Need clarifications.

Response: Dear reviewer, we really appreciate this nice suggestions. orthologous gene pairs does not represent orthologous genes, it represent the syntenic relationships betweent orthologous genes. A synteny analysis among HXKs from G.max, P.vulgaris, O.sativa, A.thaliana, M.truncatula, and S.bicolor was performed in the present study to gain some insight into the potential function of GmHXKs. As shown in Fig. 3A, the seventeen GmHXKs were scattered along ten out of twenty soybean chromosomes and each of the ten chromosomes comprised one to three GmHXKs. Moreover, a total of 30 orthologous pairs of HXKs were found in the above six species (Fig. 3A, Table S3). The GmHXKs had syntenic relationship only with PvHXKs, AtHXKs, SbHXKs and MtHXKs, including four orthologous gene pairs between G.max and S.bicolor, 11 orthologous gene pairs between G.max and A.thaliana, 16 orthologous gene pairs between G.max and M.truncatula, and 22 orthologous gene pairs between G.max and P.vulgaris (Fig. 3A). Besides, 24 paralogous HXK gene pairs were confirmed in soybean genome, and the paralogous gene pairs were apt to be found among the members in the same subfamily (Fig. 3A). These synteny occurrences proved that many MDH genes had already evolved before the divergence of soybean species. As you advised, we have realized that the description about the syntenic relationships were not clear. Therefore, we have revised some sentences or descriptions in the newly submitted manuscript to make it more accurate. The changes were highlighted in yellow (Results section, lines 271-283).

5.Ln 394-395: Remove () from GmHXK1-7, 10, and 12-16.Ln402: Now there are 51 GmHKXs?

Response: 1) We thank the reviewer for the correction and the modification has been made in “Discussion section” (Discussion section, lines 441-442). 2) Sorry about this unclarity, six plants were analyzed phylogenetically for their 51 HXK proteins; GmHXKs displayed a particular similarity for HXKs from P. vulgaris and M. truncatula of leguminous species. There are 51 HXK proteins in six plant species. Also, to be more clear and in accordance with the reviewer concerns, we have revised this sentence to make it more accurate (Discussion section, lines 449-451).

Reviewer 3 Report

Manuscript describes the identification of the Hexokinase (HXK) gene family of soybean.  The manuscript describes the exonic structure and promoter region of each family member. The manuscript then describes the expression pattern of each HXK using an expression atlas from JGI. The manuscript also presents actual biochemical activity measurements for particular plant structures.

1.     Lines 108-109: What program was used at expasy.org? What parameters were used?

2.     Lines 113-115: What parameters were used for TargetP?  What parameters were used for TopPred2?

3.     Lines 116-120: What parameters were used for the MEGA alignment? What is the location (URL) of the Plant Genome Duplication Database (PGDD)?  What parameters were used if any?

4.     Lines 124-127: What was used as the input sequence for the PlantCARE database search?

5.     Lines 130-131: Why did you decide to use the root expression as a “calibrator”? Describe what a “calibrator” is and how it is used, in brief.

6.     Line 132: What is the “second stage of trifoliate”? What ontology are you using and what is the accession number?

7.     Lines 132-133: What is the Sodium Carbonate and PEG/Mannitol solutions supposed to do? You should explain what they are being used for.

8.     Lines 139-143: the HXK assay, what procedure did you use?  Was it as per the manufacturers protocol or your own custom protocol?

9.     Lines 145-146: You should have a map of the vector and a restriction map of GmHXK7 CDS. This is to show where those NcoI and XhoI sites are in relationship to the exons.

10.  Lines 154-156: Explain How the parameters were experimentally derived.  What kit did you use, what instrument was used.  You had to use something to generate the data for the Eadie-Hofstee plot.

11.  Lines 160-161: You have to have a vector map and where and how you inserted the insert into the vector.

12.  Lines 162-163: You have to cite and briefly describe the transformation procedure you used. Saying  ”polyethylene glycol (PEG)-mediated protoplast transformation technique.” Is not sufficient to repeat the procedure.

13.  Lines 168-169: I am not sure if a new transfection vector was made to transform K599.  Describe this more fully.

14.  Lines 180-184: How do these parameters match with the HXK’s from Arabidopsis? Compare and contrast with each HXK and the At Homolog (Fig. 1A).

15.  Lines 215-216: If your GmHXK8 is truncated, how do you know if it functions as a Hexokinase at all?  You have an assay, why did you not check it?

16.  Lines 227-228: How can you infer function from synteny? If you can not explain this, remove from text.

17.  Lines 276-277: You did not observe expression of Your GmHXK3 in any tissues examined.  Are you sure it is not a pseudogene?  Did you look for any reason that this gene was not expressed??

18.  Lines 338-346:  Fig7b, did you determine the Hexokinase activity of each GmHXK?  I don’t think I saw that in the M&M section. I saw that you measured the total Hexokinase activity of different tissues, but did not see how you measured the activity of each HXK.

19.  Lines 347-374: Why did you focus on HXK7? You should explain.

There is a problem. There is an earlier manuscript that covers the GmHXK genes and it was just published. (https://www.frontiersin.org/articles/10.3389/fgene.2023.1135290/full) The authors (Chen, Tian and Guo) appear to have covered a lot of the same ground.  Because of this publication, the gene names used in this manuscript (plants-2503916) will have to be harmonized with those names. 

There were a lot of partial sentences. Some sentences missing verbs or subjects.

Author Response

Reviewer #3:

Manuscript describes the identification of the Hexokinase (HXK) gene family of soybean. The manuscript describes the exonic structure and promoter region of each family member. The manuscript then describes the expression pattern of each HXK using an expression atlas from JGI. The manuscript also presents actual biochemical activity measurements for particular plant structures.

Response: Thank you very much for your time on reviewing our manuscript. We have carefully read the thoughtful comments from you and found that these suggestions are very helpful for us to improve our manuscript. In addition, a careful revision on the manuscript was done as suggested, and the explanation of what we have changed in response to your concerns is listed below.

Major comments: 

  1. Lines 108-109: What program was used at expasy.org? What parameters were used?

Response: We thank the reviewer for bring out this very important question to improve our manuscript. The physicochemical properties of HXK proteins were analyzed by using ExPASys ProtParam tool with defalt parameters. As you advised, in the newly submitted manuscript, we have added more details about the program used at expasy.org (Materials and methods section, lines 121-122).

  1. Lines 113-115: What parameters were used for TargetP? What parameters were used for TopPred2?

Response: We thank the reviewer for pointing out this question. Protein localization was predicted using the TargetP or TopPred2 server with defalt parameters. We just pick the plant version for sequences from higher plants. As you suggestions, a more detailed description about the prediction of protein localization have been provided in the newly submitted manuscript (Materials and methods section, lines 125-127 ).

  1. Lines 116-120: What parameters were used for the MEGA alignment? What is the location (URL) of the Plant Genome Duplication Database (PGDD)? What parameters were used if any?

Response: We thank the reviewer for pointing out this question. Sorry about this unclarity, full-length HXKs from G. max (GmHXKs), Z. mays (ZmHXKs), O. sativa (OsHXKs), Phaseolus vulgaris (PvHXKs), Medicago truncatula (MtHXKs), Sorghum bicolor (SbHXKs), Brachypodium distachyon were utilized to construct a neighbor-joining phylogenetic tree by MEGA 5.0 software with the bootstrap values performed on 1000 replicates (see S2 Table for the information of these genes). In the newly submitted manuscript, a clearer description about the construction of phylogenetic tree has been provided in the Materials and Methods section “Analyses of Evolutionary, gene structure, and synteny of GmHXKs” , and and the relative reference was also added (Materials and methods section, lines 127-129). 2) Sorry about this unclarity, the location (URL) of the Plant Genome Duplication Database (PGDD) has been provided in the revised manuscript. The syntenic blocks among HXKs from different plants were identified from the plant genome duplication database (PGDD, http://chibba. agtec.uga.edu/duplication/) with defalt parameters (Materials and methods section, lines 132-134 ).

  1. Lines 124-127: What was used as the input sequence for the PlantCARE database search?

Response: We appreciate the reviewer pointing out this misstatement. We are very sorry for this negligence in the previous version of our manuscript, and 2.0 kb upstream of the position of the ATG codon in these genes were obtained from the soybean genetics and genomics database (https://www.soybase.org/sbt) to investigate the critical cis-acting elements in the promoter of GmHXK genes. A clearer description about the methods of promoter analysis of GmHXKs has been provided in the newly submitted manuscript (Materials and methods section, lines 138-140 ).

  1. Lines 130-131: Why did you decide to use the root expression as a “calibrator”? Describe what a “calibrator” is and how it is used, in brief.

Response: To examine the transcriptional profiling of GmHXKs under various abiotic stresses, soybean seedlings at the second trifoliolate stage were subjected to salt stress induced by 120 mM NaCl, alkali stress induced by 100 mM NaHCO3, and osmotic stress induced by 200 mM mannitol solutions. The total RNA from root samples at 0 h, 3h, 6 h, 12 h and 24h after treatment was isolated using a Quick Total RNA Isolation Kit (HUAYUEYANG, Beijing, China). The transcripts of GmHXKs in soybean roots under normal environment condition (untreated) were used as a calibrator. GmGAPDH and GmACTIN were used as internal reference. Each quantitative real time-polymerase chain reaction (qRT-PCR) reaction was performed in triplicate (technical replicates) on three biological replicates. The relative expression of genes was calculated by the ∆∆CT method: ΔΔCT = (CT(target, untreated) − CT(ref,untreated)) − (CT(target,treated) −CT(ref, treated)); fold-change = 2^(−∆∆CT). We are very sorry for this negligence in the previous version of our manuscript, and a brief description has been added in the revised manuscript (Materials and methods section, lines 141-151).

  1. Line 132: What is the “second stage of trifoliate”? What ontology are you using and what is the accession number?

Response: Thank you very much for your help on pointing out our problems. Soybean growth stages begin with the emergence of cotyledons from the soil surface (VE). When the unifoliate leaves unfold, the plant has reached the VC stage. When the first trifoliate leaves are fully expanded, numbers are used to signify each vegetative (V) and reproductive (R) stage of growth. When the plant begins to set flowers, the growth stages become reproductive and the plant progresses through pod development, seed development, and plant maturity. Vegetative growth stages begin to overlap reproductive stages at about R1. A new growth stage is established when 50% or more of the plants meet the requirements of the growth stage.The second stage of trifoliate is corresponding to the V2 stage, and its accession number is SOY:0000017 in Soybean Ontologies database (https://www.soybase.org/ontology.php). During the V2 stage, the second trifoliate leaf is established, and root nodules begin to develop (Fehr and Caviness, 1977). As you advised, we have added a detailed description about the “second stage of trifoliate” (Materials and methods section, lines 107-108).

  1. Lines 132-133: What is the Sodium Carbonate and PEG/Mannitol solutions supposed to do? You should explain what they are being used for.

Response: Sorry about this unclarity, we did not provide much more info at the beginning because the word limitation of the manuscript. NaCl and NaHCO3 solutions  was used to simulate salt stress and alkali stress treatments, respectively. Mannitol was used to provoke osmotic stress, to mimic the physiological condition during the drought stress. Also, to be more clear and in accordance with the reviewer concerns, we have added a brief description in Methods section to explain what they are being used for (Materials and methods section, lines 107-110).

  1. Lines 139-143: the HXK assay, what procedure did you use? Was it as per the manufacturers protocol or your own custom protocol?

Response: Thank you very much for your help on pointing out our problems. HXK activity under different abiotic stress was measured using a HXK assay kit (Solarbio Science and Technology, Beijing, China), according to the manufacturer’s protocol. As you advised, a clearer description has been added in the revised manuscript (Materials and methods section, lines 159-161).

  1. Lines 145-146: You should have a map of the vector and a restriction map of GmHXK7 CDS. This is to show where those NcoI and XhoI sites are in relationship to the exons.

Response: We thank the reviewer for this nice suggestion. As you advised, the map of a plasmid vector with restriction enzyme cutting site of GmHXK7 CDS have been added as supporting information file, Figure S2 (Materials and methods section, line 166; Supporting information section, line 25).

  1. Lines 154-156: Explain How the parameters were experimentally derived. What kit did you use, what instrument was used. You had to use something to generate the data for the Eadie-Hofstee plot.

Response: We thank the reviewer for pointing out this question. Sorry about this unclarity, determination of Km values for fructose and glucose and the hexose phosphorylation activity of GmHXK7 was performed using a HXK assay kit (Solarbio Science and Technology). As you advised, in the newly submitted manuscript, we have added more details about the approach used to calculate the kinetic parameters of GmMDH2 (Materials and methods section, lines 174-178).

  1. Lines 160-161: You have to have a vector map and where and how you inserted the insert into the vector.

Response: We thank the reviewer for this nice suggestion. We have supplemented the vector map with restriction enzyme cutting site in the newly submitted supporting information file, Figure S3 (Materials and methods section, line 183; Supporting information section, line 27).

  1. Lines 162-163: You have to cite and briefly describe the transformation procedure you used. Saying ”polyethylene glycol (PEG)-mediated protoplast transformation technique.” Is not sufficient to repeat the procedure.

Response: Thank you very much for pointing out our problems. Sorry for our careless mistake, and a more detailed description of the methodology about the transformation procedure has been provided in the newly submitted manuscript (Materials and methods section, lines 184-191).

  1. Lines 168-169: I am not sure if a new transfection vector was made to transform K599. Describe this more fully.

Response: We thank the reviewer for bring up this question. The plasmid of pBI121-GmHXK7::GFP was transformed by electroporation into Agrobacterium rhizogenes strain K599, which was used to transform soybean hypocotyls. Soybean transformation in “SN14” hypocotyls and hairy root induction were performed as reported previously (Tóth et al., 2016). Soybean plants infected with the A. rhizogenes strain K599 were considered as control hairy roots. In the newly submitted manuscript, a clearer description about the transformation of Agrobacterium rhizogenes K599 has been provided in the Materials and Methods section (Materials and methods section, lines 194-199).

  1. Lines 180-184: How do these parameters match with the HXK’s from Arabidopsis? Compare and contrast with each HXK and the At Homolog (Fig. 1A).

Response: Dear reviewer, we really appreciate this nice suggestions. In the newly submitted manuscript, we have supplemented these parameters of HXK from Arabidopsis (Table S1). Meanwhile, we have added some descriptions about the comparison of the physicochemical properties of HXK proteins between soybean and Arabidopsis, and the relative reference was also updated (Results section, lines 210-222).

  1. Lines 215-216: If your GmHXK8 is truncated, how do you know if it functions as a Hexokinase at all? You have an assay, why did you not check it?

Response: We thank the reviewer for pointing out this question. GmHXK8 has the same essential and distinct protein domains (PF00349 and PF03727) as other typical HXKs. Also, in an earlier manuscript (Chen et al., 2023) that covers the GmHXK genes which also contained the GmHXK8 (corresponding to GmHXK2, Glyma.09G144600.1). The paper proved GmHXK8 functions as a Hexokinase. In our manuscript, we are just focus on the key gene which powerfully respond abiotic stress, so we did not check it.  

  1. Lines 227-228: How can you infer function from synteny? If you can not explain this, remove from text.

Response: Thank you very much for your help on pointing out our problems. Figure 3 showed the syntenic analysis of HXK genes from different plant species. During these analyses, we found that GmHXKs showed collinear relations with HXKs from M. truncatula, P.vulgaris, and A. thaliana, suggesting that GmHXKs may have arisen prior to the divergence of the Arabidopsis, lucerne grass and common bean lineages (Fig. 3). And as your suggestions, a clearer description about the results of the syntenic analysis has been provided in the newly submitted manuscript (Results section, lines 272-284).

  1. Lines 276-277: You did not observe expression of Your GmHXK3 in any tissues examined. Are you sure it is not a pseudogene? Did you look for any reason that this gene was not expressed??

Response: We thank the reviewer for bring up this question. We evaluated the transcriptional patterns of GmHXKs in multiple tissues via high-throughput sequencing data from Phytozome database, including leaves, roots, seeds, pods, and flowers and so on. The findings are shown in the form of heat maps. We recheck the data and found that GmHXK3 is only expressed in flower, and the expression data is very low, so it didn't show up on the heat map. Additionally, to be more clear and in accordance with the reviewer concerns, the figure has been revised, and we have revised some sentences or descriptions in the newly submitted manuscript to make it more accurate. The changes were highlighted in yellow (Results section, lines 315-325).

  1. Lines 338-346: Fig5b, did you determine the Hexokinase activity of each GmHXK? I don’t think I saw that in the M&M section. I saw that you measured the total Hexokinase activity of different tissues, but did not see how you measured the activity of each HXK.

Response: We thank the reviewer for pointing out this question. Fig5b represents the correlation coefficient between HXK enzymatic activity and the expression levels of each GmHXK. We did not determine the Hexokinase activity of each GmHXK, we just measured the total Hexokinase activity under various abiotic stresses.

  1. Lines 347-374: Why did you focus on HXK7? You should explain.

Response: Thank you for your nice suggestions on improving our manuscript.  GmHXK7 have been selected for genetic validation due to its higher levels under alkali stress. GmHXK7 respond more vigorously to alkali stress than other genes. The average mRNA abundances of GmHXK7 under alkali stress was much higher than that of other genes, indicating that it may be a particularly important regulator of salines-alkali stress tolerance. Besides, correlation analysis revealed the trend in HXK activity under abiotic stresses was well consistent with gene expression of GmHXK7, which indicated that it to encode the major HXK isoform involved in response to abiotic stresses. As you suggested, we have added some description about the reasons. Changes have been made and annotated in the article (Results section, lines 381-384).

  1. There is a problem. There is an earlier manuscript that covers the GmHXK genes and it was just published. (https://www.frontiersin.org/articles/10.3389/fgene. 2023. 1135290/full) The authors (Chen, Tian and Guo) appear to have covered a lot of the same ground. Because of this publication, the gene names used in this manuscript (plants-2503916) will have to be harmonized with those names. Response: Thanks again for the reminder; as your suggestion, a careful correction has been made on all the Figs and the whole manuscript to make sure that the gene names used in this manuscript (plants-2503916) harmonized with those names in that manuscript (Chen et al., 2023) (Abstract section, lines 9, lines 11, lines 13, lines 16, lines 17-20; Introduction section, lines 89-94; Materials and Methods section, lines 165, lines 168, line 179-180; Results section, lines 214-215, lines 226, lines 228, lines 247, lines 250, lines 261, lines 263-265, lines 287, lines 302-306, lines 329, lines 331, lines 333-335, lines 358, lines363, lines 365-366, lines 395-397, lines 404-405, lines 407-408, lines 410, lines 414-417, lines 422, lines 424-427, lines 429; Discussion section, lines 440, lines 442-443, lines 447, lines 466-467, lines 483-485, lines 500, lines 506, lines 510, lines 512, lines 514, lines 522, lines 524; Conclusions section, lines 533, lines 535). The corresponding relationships are as follows:

Reviewer 4 Report

The title of the paper is not appropriate. The introduction is too long and containing many non relevant information. Phytomaterials is not an appropriate word. Lack of information regarding Methods and Plant material, growth conditions, RNA extraction protocol, what are independent technical analysis? No information about statistical analysis, except when describing Fig 5 - 8). Results are not clear if we take into consideration described methods. There is no clear goal of the paper.

Major revisions.

Author Response

Reviewer #4:

The title of the paper is not appropriate. The introduction is too long and containing many non relevant information. Phytomaterials is not an appropriate word. Lack of information regarding Methods and Plant material, growth conditions, RNA extraction protocol, what are independent technical analysis? No information about statistical analysis, except when describing Fig (5-8). Results are not clear if we take into consideration described methods. There is no clear goal of the paper.

Response: Thank you very much for your nice suggestions on improving our manuscript. 1) We have made correction in the title according to your comment, and the revised manuscript has been submitted online (Title section, lines 1-4). 2) As you advised, we have realized that some sentences or descriptions in the Introduction section were not appropriate in the previous manuscript, and a careful modification has been made in the revised manuscript (Introduction section, lines 57-65, lines 70-73, lines 77-84). 3) Thank you very much for bring out this problem. The word “Phytomaterials” has been revised in the newly submitted manuscript (Materials and methods section, line 98). 4) Sorry about this unclarity, we did not provide much more info at the beginning because the word limitation of the manuscript. As you suggestions, a more detailed description of the methodology has been provided in the newly submitted manuscript and the relative reference has also been added (Materials and methods section, lines 98-112, lines 121-122, lines 125-132, lines 136-151, lines 154-161, lines 166-177, lines 183-207). 5) Sorry about this unclarity, in the newly submitted manuscript, a clearer description about statistical analysis methods has been provided in the Materials and Methods section (Materials and methods section, lines 204-207). We thank the reviewer for giving us the opportunities for revising and improving our manuscript. We had tried our best to explore the functions of GmHXKs in the development and stress adaption of soybean, and we hope these studies could provide some useful information and derive some biological hypothesis on the function of HXK members in plants.

Round 2

Reviewer 3 Report

The Background, discussion and reference sections are incomplete.  There is no mention of the Chen et al. 2023 paper in those sections.  It does appear that the gene symbols have been harmonized with that paper which is good, but it needs to be included in the background,  discussion and reference section of the paper.  It might also be good to mention any differences from that paper ie genes identified in this study that were not identified in the previous paper and vise-a-versa.  If there are differences, it would be good to put in the discussion section an analysis of how the two procedures differ and how that lead to the different results.

Author Response

Dear reviewer, we really appreciate this nice suggestions. As you advised, a careful modification has been made in the revised manuscript, and a more detailed description about the results of the Chen et al. 2023 paper have been provided in the newly submitted manuscript (Introduction section, line 37, line 40, lines 70-75; Discussion section, lines 432-434, lines 463-465, lines 496-498), and the relative reference was also updated (Reference section, lines 598-599).